



**Bimodal or quadrimodal? Statistical tests for the shape of fault patterns**
David Healy[1*] & Peter Jupp[2]
[1]School of Geosciences, King's College, University of Aberdeen, Aberdeen AB24 3UE Scotland
[2]School of Mathematics & Statistics, University of St Andrews, St Andrews KY16 9SS Scotland
*Corresponding author e-mail: d.healy@abdn.ac.uk
**Abstract**
Natural fault patterns, formed in response to a single tectonic event, often display significant
variation in their orientation distribution. The cause of this variation is the subject of some
debate: it could be 'noise' on underlying conjugate (or bimodal) fault patterns or it could be
intrinsic 'signal' from an underlying polymodal (e.g. quadrimodal) pattern. In this
contribution, we present new statistical tests to assess the probability of a fault pattern
having two (bimodal, or conjugate) or four (quadrimodal) underlying modes. We use the
eigenvalues of the 2nd and 4th rank orientation tensors, derived from the direction cosines of
the poles to the fault planes, as the basis for our tests. Using a combination of the existing
fabric eigenvalue (or modified Flinn) plot and our new tests, we can discriminate reliably
between bimodal (conjugate) and quadrimodal fault patterns. We validate our tests using
synthetic fault orientation datasets constructed from multimodal Watson distributions, and
then assess six natural fault datasets from outcrops and earthquake focal plane solutions. We
show that five out of six of these natural datasets are probably quadrimodal. The tests have
been implemented in the R language and a link is given to the authors' source code.

**1. Introduction**
*1.1 Background*
Faults are common structures in the Earth's crust, and they rarely occur in isolation. Patterns
of faults, and other fractures such as joints and veins, control the bulk transport and
mechanical properties of the crust. For example, arrays of low permeability (or 'sealing')
faults in a rock matrix of higher permeability can produce anisotropy of permeability and
preferred directions of fluid flow. Arrays of weak faults can similarly produce anisotropy – i.e.
directional variations – of bulk strength. It is important to understand fault patterns, and
quantifying the geometrical attributes of any pattern is an important first step. Faults, taken
as a class of brittle shear fractures, are often assumed to form in conjugate arrays, with fault
planes more or less evenly distributed about the largest principal compressive stress, $\sigma_1$, and
making an acute angle with it. This model, an amalgam of theory and empirical observation,
predicts that conjugate fault planes intersect along the line of $\sigma_2$ (the intermediate principal
stress) and the fault pattern overall displays bimodal symmetry (Figure 1a). A fundamental
limitation of this model is that these fault patterns can only ever produce a plane strain
(intermediate principal strain $\varepsilon_2 = 0$), with no extension or shortening in the direction of $\sigma_2$.



This kinematic limitation is inconsistent with field and laboratory observations that
document the existence of polymodal or quadrimodal fault patterns, and which produce
triaxial strains in response to triaxial stresses (e.g. Aydin & Reches, 1982; Reches, 1978;
Blenkinsop, 2008; Healy et al., 2015; McCormack & McClay, 2018). Polymodal and
quadrimodal fault patterns possess orthorhombic symmetry (Figure 1b & 1c).

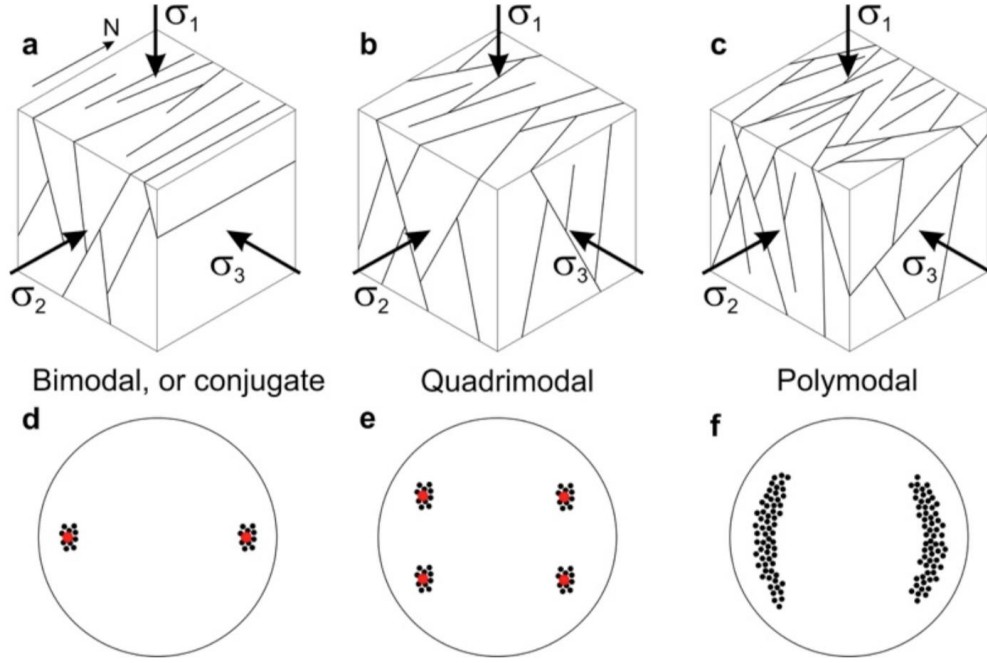


**Figure 1.** Schematic diagrams to compare conjugate fault patterns displaying bimodal
symmetry with quadrimodal and polymodal fault patterns displaying orthorhombic
symmetry. **a-c**) Block diagrams showing patterns of normal faults and their relationship to
the principal stresses. **d-f**) Stereographic projections (equal area, lower hemisphere) showing
poles to fault planes for the models shown in a-c. Natural examples of all three patterns have
been found in naturally deformed rocks.

Fault patterns are most often visualised through maps of their traces and equal-angle
(stereographic) or equal-area projections of poles to fault planes or great circles. Azimuthal
projection methods (hereafter 'stereograms') provide a measure of the orientation
distribution, including the attitude and the shape of the overall pattern. However, these plots
can be unsatisfactory when they contain many data points, or the data are quite widely
dispersed. Woodcock (1977) developed the idea of the fabric shape, based on the fabric or
orientation tensor of Scheidegger (1965). The eigenvalues of this $2^{nd}$ rank tensor can be used
in a modified Flinn plot (Flinn, 1962; Ramsay, 1967) to discriminate between clusters and
girdles of poles. These plots can be useful for three of the five possible fabric symmetry
classes – spherical, axial and orthorhombic – because the three fabric eigenvectors coincide
with the three symmetry axes. However, there are issues with the interpretation of
distributions that are not uniaxial (Woodcock, 1977). We address these issues in this paper.



Reches (Reches, 1978; Aydin & Reches, 1982; Reches, 1983; Reches & Dieterich, 1983) has
exploited the orthorhombic symmetry of measured quadrimodal fault patterns to explore the
relationship between their geometric/ kinematic attributes and tectonic stress. More recently,
Yielding (2016) measured the branch lines of intersecting normal faults from seismic
reflection data and found they aligned with the bulk extension direction – a feature consistent
with their formation as polymodal patterns. Bimodal (i.e. conjugate) fault arrays have branch
lines aligned perpendicular to the bulk extension direction.
*1.2 Rationale*
The fundamental underlying differences in the symmetries of the two kinds of fault pattern –
bimodal/bilateral and polymodal/orthorhombic – suggest that we should test for this
symmetry using the orientation distributions of measured fault planes. The results of such
tests may provide further insight into the kinematics and/or dynamics of the fault-forming
process. This paper describes new tests for fault pattern orientation data, and includes the
program code for each test written in the R language (R Core Team, 2017). The paper is
organised as follows: the next section (2) reviews the kinematic and mechanical issues raised
by conjugate and polymodal fault patterns, and in particular, the implications for their
orientation distributions. Section 3 describes the datasets used in this study, including
synthetic and natural fault orientation distributions. Section 4 presents tests for assessing
whether an orientation distribution has orthorhombic symmetry, including a description of
the mathematics and the R code.  The examples used include synthetic orientation datasets of
known attributes (with and without added 'noise') and natural datasets of fault patterns
measured in a range of rock types.  A Discussion of issues raised is provided in Section 5, and
is followed by a short Summary. The R code is available from http://www.mcs.st-
andrews.ac.uk/~pej/2mode_tests_Rcode190418.

## 2. Bimodal (conjugate) versus quadrimodal fault patterns

Conjugate fault patterns should display bimodal or bilateral symmetry in their orientation
distributions on a stereogram, and ideally show evidence of central tendency about these two
clusters (Figure 1d; Healy et al., 2015). Quadrimodal fault patterns should show orthorhombic
symmetry and, ideally, evidence of central tendency about the four clusters of poles on
stereograms (Figure 1e). More general polymodal patterns should show orthorhombic
symmetry with an even distribution of poles in two arcs (Figure 1f). For data collected from
natural fault planes some degree of intrinsic variation, or 'noise', is to be expected. Two
natural example datasets are shown in Figure 2. The Gruinard dataset is from a small area (~
5 m$^2$) in one outcrop of Triassic sandstone, and shows poles to deformation bands with small
normal offsets (mm-cm). The Flamborough dataset is taken from Peacock & Sanderson (1992;
their Figure 2a) and shows poles to normal faults in the Cretaceous chalk along a coastline
section of about 1.8 km. The authors clearly state that the approximately E-W orientation of
the coastline may have generated a sampling bias in the measured data (i.e. a relative under-
representation of E-W oriented fault planes). Both datasets illustrate the nature of the
problem addressed in this paper: given variable, incomplete and noisy data of different
sample sizes, how can we assess the symmetry of the underlying fault pattern?





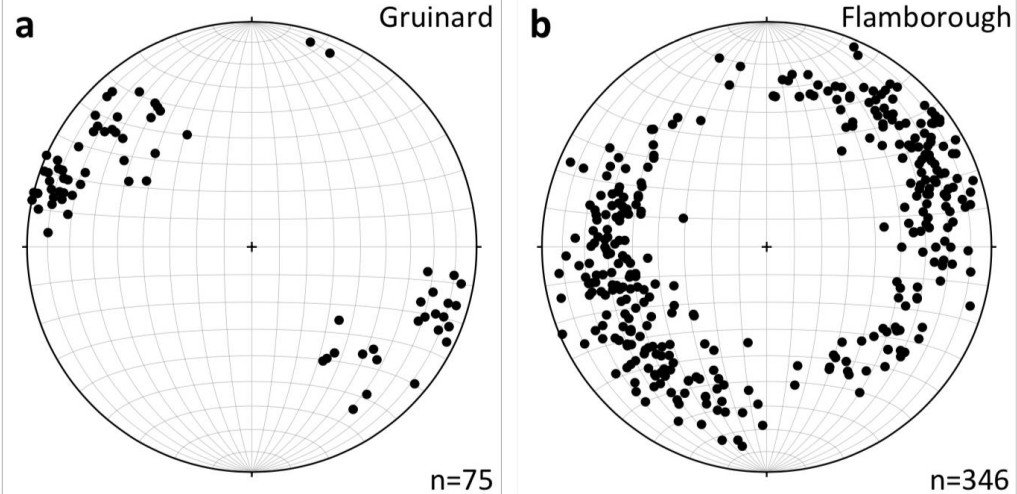


**Figure 2.** Stereographic projections (equal area, lower hemisphere) showing two natural fault datasets. a) Poles to deformation bands (small offset faults; *n*=75) measured in Triassic sandstones at Gruinard Bay, NW Scotland (Healy et al., 2006a, b). These data were collected from a small contiguous outcrop, approximately 10 m² in area. b) Poles to faults measured in Cretaceous chalk at Flamborough Head, NE England (*n*=346). These data have been taken from a figure published in Peacock & Sanderson (1992) and re-plotted in the same format as those from Gruinard.

## 3. Datasets used in this study

### 3.1. Synthetic datasets

We use two sets of synthetic data to test our new statistical methods, both based on the Watson orientation distribution (Fisher et al., 1987 section 4.4.4; Mardia & Jupp, 2000 section 9.4.2). This is the simplest non-uniform distribution for describing undirected lines, and has probability density

$$f(\pm x; \mu, \kappa) \propto exp\{\kappa(\mu^T x)^2\}$$

where $\kappa$ is a measure of concentration (low $\kappa$ = dispersed, high $\kappa$ = concentrated) and $\mu$ is the mean direction. To obtain a synthetic conjugate fault pattern dataset of size *n* we combined two datasets of size *n*/2, each from a Watson distribution, the two mean directions being separated by 60°. We generated synthetic bimodal datasets with $\kappa$ = 10, 20, 50 and 100 and *n*=52 and 360 (Figure 3). This variation in $\kappa$ provides a useful range of concentrations encompassing those observed in measured natural data, and can be considered as a measure of 'noise' within the distribution. Many natural datasets are often small due to limitations of outcrop size, and the two sizes of synthetic distribution (*n*=52 and 360) allow for this fact. For synthetic polymodal fault patterns, we generated quadrimodal datasets of size *n* by combining four Watson distributions of size *n*/4 with their mean directions separated by 60° in dip (as



above) and 52° in strike (see Healy et al., 2006a, b). By varying $n$ from 52 to 360 we cater for
comparisons with smaller and larger natural datasets, and as for the synthetic bimodal
datasets, we varied $\kappa$ in the range 10, 20, 50 and 100 (Figure 4).

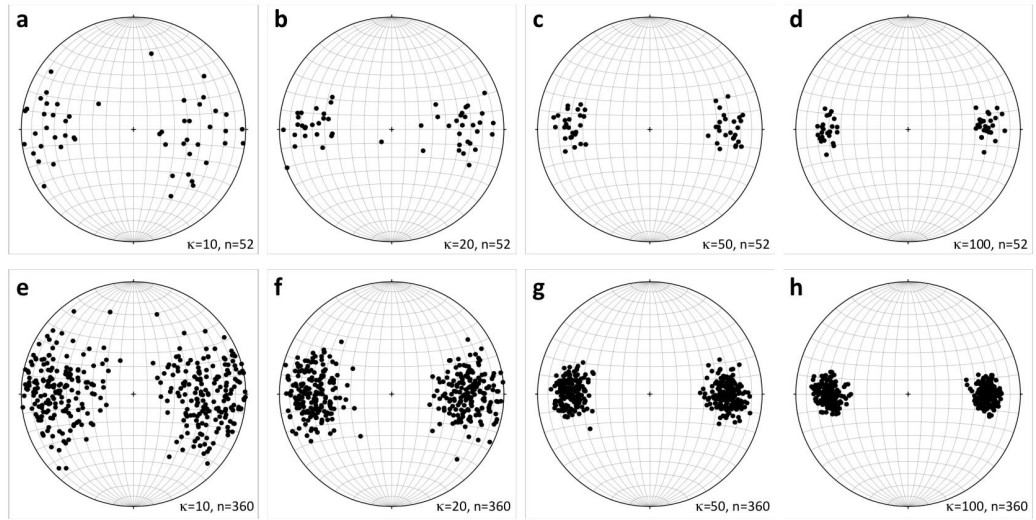


**Figure 3.** Stereographic projections (equal area, lower hemisphere) showing the eight
synthetic datasets designed to model conjugate (bimodal) fault patterns in this study. **a-d)**
Synthetic fault datasets derived from equal mixtures of two Watson distributions with mean
pole directions separated by an inter-fault dip angle of 60 degrees. These models represent a
'low fault count' scenario, with $n$ = 52 and $\kappa$ (the Watson dispersion parameter) varying from
10 to 100. **e-h)** These models represent a 'high fault count' scenario, with $n$ = 360 and $\kappa$
varying from 10 to 100.


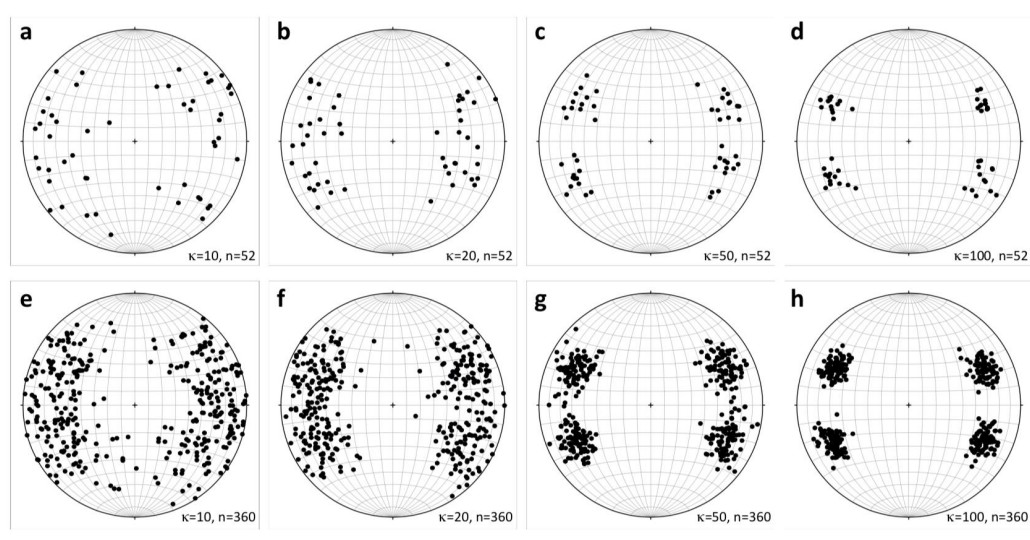






**Figure 4.** Stereographic projections (equal area, lower hemisphere) showing the eight
synthetic datasets designed to model quadrimodal fault patterns in this study. **a-d**) Synthetic
fault datasets derived from equal mixtures of four Watson distributions with mean pole
directions separated by an inter-fault dip angle of 60 degrees and a strike separation of 52
degrees. These models represent a 'low fault count' scenario, with $n$ = 52 and $\kappa$ (the Watson
dispersion parameter) varying from 10 to 100. **e-h**) These models represent a 'high fault
count' scenario, with $n$ = 360 and $\kappa$ varying from 10 to 100.

*3.2. Natural datasets*
We use six natural datasets of fault plane orientations from regions that have undergone or
are currently undergoing extension - i.e. we believe the majority of these faults display normal
kinematics (Figure 5). The Gruinard dataset (Figure 5a) is from Gruinard Bay in NW Scotland
(UK), and featured in previous publications (Healy et al., 2006a, b). The most important thing
about this dataset is that the fault planes were all measured from a small area ($\sim$5 m$^2$) of
contiguous outcrop of a single sandstone bed. This means it is highly unlikely that the
orientation data are affected by any local stress variations and subsequent possible rotations.
The data were measured in normal-offset deformation bands with displacements of a few
millimetres to centimetres. The next three datasets have been digitised from published papers
on normal faults in Utah (Figure 5b; Chimney Rock; Krantz, 1989), northern England (Figure
5c; Flamborough; Peacock & Sanderson, 1992) and Italy (Figure 5d; Central Italy; Roberts,
2007). In each case, the published stereograms were digitised to extract Cartesian ($x,y$)
coordinates of the poles to faults, and these were then converted to plunge and plunge
direction using the standard equations for the projection used (e.g. Lisle & Leyshon, 2004).
Slight differences in the number of data plotted for each of these three with respect to the
original publication arise due to the finite resolution of the digitised image of the stereograms.
The last two datasets for the Aegean and Tibet (Figure 5e & f) are derived from earthquake
focal mechanisms using the CMT catalogue (Ekström et al., 2012). In each case the steepest
dipping nodal plane was selected in the absence of convincing evidence for low-angle normal
faulting in these regions.





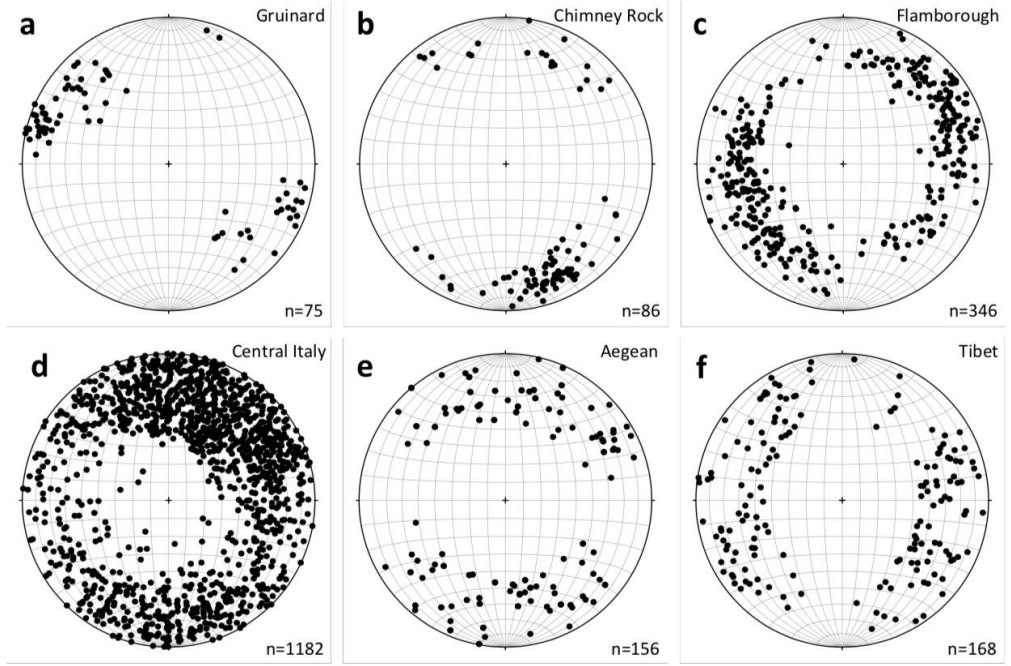

**Figure 5.** Stereographic projections (equal area, lower hemisphere) showing the six natural datasets used in this study. All plots show poles to faults, the majority of which are inferred to be normal. **a**) Data from deformation bands measured in faulted Triassic sandstones at Gruinard Bay, Scotland (Healy et al., 2006a; 2006b). **b**) Data from faults and measured in sandstones at Chimney Rock in the San Rafael Swell, Utah, USA. Data digitised from Krantz (1989). **c**) Data from faults measured in cliffs of Cretaceous chalk at Flamborough Head, NE England. Data digitised from Peacock & Sanderson (1992). **d**) Data from faults measured in the Apennines of Central Italy. Data digitised from Roberts (2007). **e**) Data from focal mechanism nodal planes derived from the CMT catalogue for the Aegean region (Ekström et al., 2012). **f**) Data from focal mechanism nodal planes derived from the CMT catalogue for the Tibet region (Ekström et al., 2012).

## 4. Testing for orthorhombicity

*4.1 Eigenvalue fabric (modified Flinn) plots*

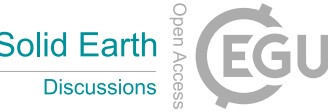

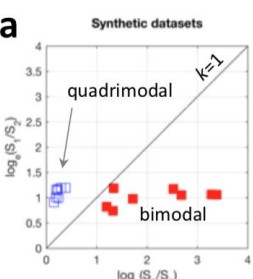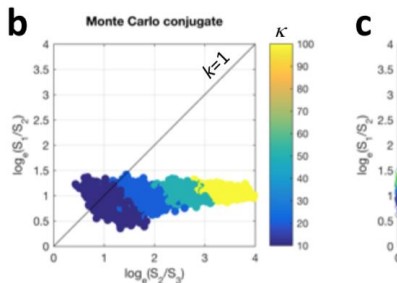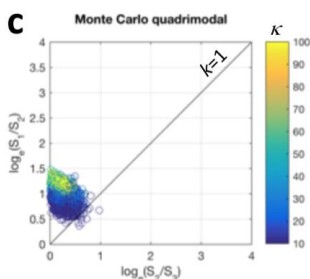

188

**Figure 6.** Graphs showing the ratios of eigenvalues of the orientation matrices for the synthetic datasets (Flinn, 1962; Ramsay, 1967; Woodcock, 1977). **a**) Synthetic conjugate (i.e. bimodal; filled red symbols) and quadrimodal (hollow blue symbols) fault data. Note that the conjugate and quadrimodal data lie either side of the line $k = 1$, where $k = \log_e(S_1/S_2)/\log_e(S_2/S_3)$. **b**) Eigenvalue ratios from a Monte Carlo simulation of conjugate fault orientations using the two Watson mixture model. 1000 simulations were run for each of four different $\kappa$ values (10, 20, 50 and 100; a total of 4000 data points), corresponding to the range of the discrete datasets shown in a). **c**) Eigenvalue ratios from a Monte Carlo simulation of quadrimodal fault orientations using the four Watson mixture model. 1000 simulations were run for each of four different $\kappa$ values (10, 20, 50 and 100; a total of 4000 data points), corresponding to the range of the discrete datasets shown in a).

We calculated the $2^{nd}$ rank orientation tensor (Woodcock, 1977) for each of the synthetic datasets shown in Figures 3 and 4 (bimodal and quadrimodal, respectively). The eigenvalues of this tensor ($S_1$, $S_2$ and $S_3$, where $S_1$ is the largest and $S_3$ is the smallest) are used to plot the data on a modified Flinn diagram (Figure 6), with $\log_e(S_2/S_3)$ on the x-axis and $\log_e(S_1/S_2)$ on the y-axis. The points corresponding to the bimodal (shown in red) and quadrimodal (shown in blue) datasets lie in distinct areas. Bimodal (conjugate) fault patterns lie below the 1:1 line, on which $S_1/S_2 = S_2/S_3$. This is due to the $S_3$ eigenvalue being very low (near 0) for these distributions, which for high values of $\kappa$ begin to resemble girdle fabric patterns confined to the plane of the eigenvectors corresponding to eigenvalues $S_1$ and $S_2$ (Woodcock, 1977). In contrast, the quadrimodal patterns lie above the 1:1 line, as $S_3$ for these distributions is large relative to the equivalent bimodal pattern (i.e. for the same values of $\kappa$ and $n$). The modified Flinn plot therefore provides a potentially rapid and simple way to discriminate between bimodal (conjugate) and quadrimodal fault patterns. Note, however, that the spread of the bimodal patterns in Figure 6a along the x-axis is a function of the $\kappa$ value of the underlying Watson distribution, with low values of $\kappa$ – low concentration, highly dispersed – lying closer to the origin. Dispersed or noisy bimodal (conjugate) patterns may therefore lie closer to quadrimodal patterns (see Discussion below).

*4.2 Randomisation tests using 2nd and 4th rank orientation tensors*

*4.2.1 Underlying distributions*

To get a suitable general setting for our tests, we formalise the construction of the bimodal and quadrimodal datasets considered in Section 3.1. Whereas the datasets considered in



Section 3.1 necessarily have equal numbers of points around each mode, for datasets arising
from the distributions here, this is true only *on average*. The very restrictive condition of
having a Watson distribution around each mode is relaxed here to that of having a circularly-
symmetric distribution around each mode.
Suppose that axes $\pm\mathbf{x_1}, \ldots \pm\mathbf{x_n}$ are independent observations from some distribution of axes. If
the parent distribution is thought to be multi-modal then two appealing models are:
(i) The **bimodal equal mixture model** can be thought of intuitively as obtained by 'pulling

apart' a unimodal distribution into two equally strong modes angle $\alpha$ apart. More

precisely, the probability density is:

$$f_2(\pm\mathbf{x}; \{\pm\boldsymbol{\mu_1}, \pm\boldsymbol{\mu_2}\}) = \tfrac{1}{2}\{g(\pm\mathbf{x}; \pm\boldsymbol{\mu_1}) + g(\pm\mathbf{x}; \pm\boldsymbol{\mu_2})\}, \qquad (1)$$
where $\pm\boldsymbol{\mu_1}$ and $\pm\boldsymbol{\mu_2}$ are axes angle $\alpha$ apart, and $g(.; \pm\boldsymbol{\mu})$ is the probability density function of
some axial distribution that has rotational symmetry about its mode $\pm\boldsymbol{\mu}$;
(ii) The **quadrimodal equal mixture model** can be thought of intuitively as obtained by

'pulling apart' a bimodal equal mixture distribution into two bimodal equal

mixture distributions with planes angle $\gamma$ apart, so that it has four equally strong modes.

More precisely, the probability density is:

$$f_4(\pm\mathbf{x}; \{\pm\boldsymbol{\mu_1}, \pm\boldsymbol{\mu_2}\}, \gamma) = \tfrac{1}{4}\sum_{\varepsilon,\eta} g(\pm\mathbf{x}; \pm\boldsymbol{\mu_{\varepsilon,\eta}}), \qquad (2)$$
where
$$\boldsymbol{\mu_{\epsilon,\eta}} = \check{c}(c\boldsymbol{v_1} + \epsilon s \boldsymbol{v_2}) + \eta \check{s} \boldsymbol{v_3} \qquad (3)$$
with $c = \cos(\alpha/2), s = \sin(\alpha/2), \check{c} = \cos(\gamma/2), \check{s} = \sin(\gamma/2), \cos(\alpha) = \boldsymbol{\mu_1'}\boldsymbol{\mu_2}$ and $(\epsilon, \eta)$
runs through $\{\pm 1\}^2$. If $\gamma = 0$, then (3) reduces to (2).
The problem of interest is to decide whether the parent distribution is (1) or (2).

*4.2.2 The tests*
Given axes $\pm\mathbf{x_1}, \ldots \pm\mathbf{x_n}$ we denote by $\pm\hat{\boldsymbol{v}}_1$ and $\pm\hat{\boldsymbol{v}}_3$, respectively, the largest and smallest
principal axes of the orientation tensor. $S_1$ and $S_3$ are the eigenvalues of this matrix. We can
also define
$$S_{11} = n^{-1} \sum_{i=1}^{n} (\hat{\boldsymbol{v}}_1' \mathbf{x}_i)^4, S_{33} = n^{-1} \sum_{i=1}^{n} (\hat{\boldsymbol{v}}_3' \mathbf{x}_i)^4.$$
$S_1$ and $S_2$ are the 2nd moments of $\pm\mathbf{x_1}, \ldots \pm\mathbf{x_n}$ along the 1st and 3rd principal axes, respectively,
whereas $S_{11}$ and $S_{33}$ are the 4th moments along these principal axes. Therefore, both $S_1 - S_3$
and $S_{11} - S_{33}$ are measures of anisotropy of $\pm\mathbf{x_1}, \ldots \pm\mathbf{x_n}$.
Some algebra shows that
$$T_1 - T_3 = \cos(\gamma)\{E[x^2] - E[v^2]\}, \qquad (4)$$



where $T_1$ and $T_3$ are the population versions of $S_1$ and $S_3$, respectively, and $\pm x$ and $\pm v$ are the
components of $\pm\mathbf{x}$ in the quadrimodal equal mixture model (2) along its 1st and 3rd principal
axes, respectively. Then (4) gives
$$\cos(\gamma) \approx \frac{S_1 - S_3}{E[x^2] - E[v^2]}$$
and therefore, it is sensible to:

reject bimodality for *small* values of $S_1 – S_3$.    (5)

Further algebra shows that

$T_{11} - T_{33} = \cos(\gamma)\{E[x^4] - E[v^4]\}$,    (6)

where $T_{11}$ and $T_{33}$ are the population versions of $S_{11}$ and $S_{33}$, respectively. Then (6) gives
$$\cos(\gamma) \approx \frac{S_{11} - S_{33}}{E[x^4] - E[v^4]}$$
and so, it is sensible to:

reject bimodality for *small* values of $S_{11} – S_{33}$.    (7)

The significance of tests (5) or (7) is assessed by comparing the observed value of the statistic
with the randomisation distribution. This is achieved by creating a further $B$ pseudo-samples
(for a suitable positive integer $B$), in each of which the $i$th observation is obtained from $\pm x_i$ by
rotating $\pm x_i$ about the closer of the 2 fitted modes through a uniformly distributed random
angle. The $p$-value is taken as the proportion of the $B+1$ values of the statistic that are smaller
than (or equal to) the observed value.

*4.3 Results for synthetic datasets*
Table 1 gives the $p$-values and corresponding decisions (at the 5% level) obtained by applying
the tests to some synthetic datasets simulated from the bimodal equal mixture model. Table 2
does the same for some datasets simulated from the quadrimodal equal mixture model. In
each case, both tests come to the correct conclusion.

| True number of modes | $\kappa$ | $n$ | $S_1 - S_3$ test | | $S_{11} - S_{33}$ test | |
|---|---|---|---|---|---|---|
| | | | $p$-value | # of modes | $p$-value | # of modes |
| 2 | 10 | 52 | 0.37 | 2 | 0.51 | 2 |
| 2 | 10 | 360 | 0.27 | 2 | 0.33 | 2 |
| 2 | 20 | 52 | 0.66 | 2 | 0.69 | 2 |
| 2 | 20 | 360 | 0.20 | 2 | 0.25 | 2 |
| 2 | 50 | 52 | 0.45 | 2 | 0.48 | 2 |
| 2 | 50 | 360 | 0.35 | 2 | 0.42 | 2 |
| 2 | 100 | 52 | 0.34 | 2 | 0.41 | 2 |



| 2 | | 100 | 360 | 0.60 | 2 | | 0.63 | 2 |
|---|---|-----|-----|------|---|---|------|---|


**Table 1.** *p*-values and corresponding decisions at 5% significance level of randomisation tests
of bimodality for bimodal equal mixtures of synthetic Watson distributions. *n*=total sample
size. *B*=999 further randomisation samples per data set (see text for details).

| True number of modes | | | $S_1 - S_3$ test | | $S_{11} - S_{33}$ test | |
|----------------------|-----|-----|---------|-----------|---------|-----------|
| $\kappa$ | $n$ | *p*-value | # of modes | *p*-value | # of modes |
| 4 | 10 | 52 | 0.00 | > 2 | 0.00 | > 2 |
| 4 | 10 | 360 | 0.00 | > 2 | 0.00 | > 2 |
| 4 | 20 | 52 | 0.00 | > 2 | 0.00 | > 2 |
| 4 | 20 | 360 | 0.00 | > 2 | 0.00 | > 2 |
| 4 | 50 | 52 | 0.00 | > 2 | 0.00 | > 2 |
| 4 | 50 | 360 | 0.00 | > 2 | 0.00 | > 2 |
| 4 | 100 | 52 | 0.00 | > 2 | 0.00 | > 2 |
| 4 | 100 | 360 | 0.00 | > 2 | 0.00 | > 2 |


**Table 2.** *p*-values and corresponding decisions at 5% significance level of randomisation tests
of bimodality for quadrimodal equal mixtures of Watson distributions. *n*=total sample size.
*B*=999 further randomisation samples per data set (see text for details).

*4.4 Results for natural datasets*
Table 3 gives the *p*-values and corresponding decisions (at the 5% level) obtained by applying
the tests to the natural datasets discussed in Section 3.2. For each dataset, the two tests come
to the same conclusion, which is plausible in view of Figure 5. Figure 7 shows the fabric
eigenvalue plot for these datasets.



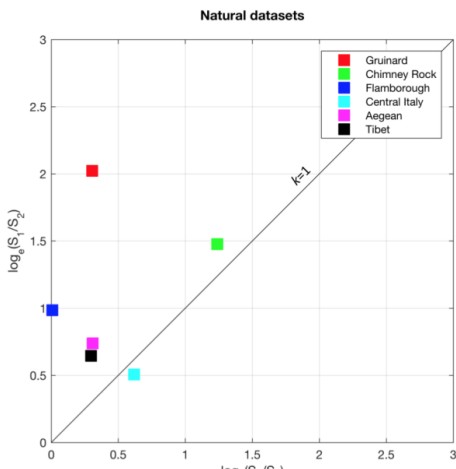


**Figure 7.** Eigenvalue ratio plot for the natural datasets shown in Figure 5. All but one dataset (Central Italy) lies above the line for $k$=1. The best-constrained quadrimodal fault dataset (Gruinard) has the highest ratio of $\log_e(S_1/S_2)$.


| Field location | | $S_1 - S_3$ test | | $S_{11} - S_{33}$ test | |
|---|---|---|---|---|---|
| | $n$ | $p$-value | # of modes | $p$-value | # of modes |
| **Gruinard** | 75 | 0.00 | > 2 | 0.00 | > 2 |
| **Chimney Rock** | 86 | 0.99 | 2 | 1.00 | 2 |
| **Flamborough** | 346 | 0.00 | > 2 | 0.00 | > 2 |
| **Central Italy** | 1182 | 0.00 | > 2 | 0.00 | > 2 |
| **Aegean** | 156 | 0.00 | > 2 | 0.00 | > 2 |
| **Tibet** | 168 | 0.00 | > 2 | 0.00 | > 2 |


**Table 3.** $p$-values and corresponding decisions at 5% significance level of randomisation tests of bimodality for natural data sets. $n$=total sample size. $B$=999 further randomisation samples per data set (see text for details).


**5. Discussion**

In the analysis described above and the tests we performed with synthetic datasets, we assumed that bimodal and quadrimodal Watson orientation distributions provide a reasonable approximation to the distributions of poles to natural fault planes. In terms of the underlying statistics this is unproven, but we know of no compelling evidence in support of alternative distributions. New data from carefully controlled laboratory experiments on rock




or analogous materials might provide important constraints for the underlying statistics of
shear fracture plane orientations.
We have tested our new methods on synthetic and natural datasets. Arguably, six natural
datasets are insufficient to establish firmly the primacy of polymodal orthorhombic fault
patterns in nature (Figure 7). However, we reiterate the key recommendation from Healy et
al. (2015): to be useful for this task, fault orientation datasets need to show clear evidence of
contemporaneity among all fault sets, through tools such as matrices of cross-cutting
relationships (Potts & Reddy, 2000). In addition, as shown above, larger datasets (n>200)
tend to show clearer patterns. Scope exists to collect fault or shear fracture orientation data
from sources other than outcrops: Yielding (2016) has measured normal faults in seismic
reflection data from the North Sea and Ghaffari et al. (2014) measured faults in cm-sized
samples deformed in the laboratory and then scanned by X-ray computerised tomography.
The Chimney Rock dataset is probably not orthorhombic according to the two tests, and lies
close to the line for $k$=1 on Figure 7. It is interesting to note that the Chimney Rock data, and
other fault patterns from the San Rafael area of Utah, are considered as displaying
orthorhombic symmetry by Krantz (1989) and Reches (1978). However, a subsequent re-
interpretation by Davatzes et al. (2003) has ascribed the fault pattern to overprinting of
earlier deformation bands by later sheared joints. This may account for the inconsistent
results of our tests when compared to the position of the pattern on the eigenvalue plot. The
Central Italy dataset (taken from Roberts, 2007) is very large ($n$=1182) and the data were
measured over a wide geographical area. The dataset lies below the line for $k$=1 on the fabric
eigenvalue plot (Figure 7), which might suggest it is bimodal. However, for fault planes
measured over large areas there is a significant chance that regional stress variations may
have produced systematically varying orientations of fault planes.
A final point concerns dispersion (noise) in the data. Synthetic datasets of bimodal
(conjugate) and quadrimodal patterns with low values of $\kappa$, the Watson concentration
parameter, fall into overlapping fields on the eigenvalue fabric plot. We ran 1000 Monte Carlo
simulations of bimodal and quadrimodal Watson distributions each with $n$=52 poles, and $\kappa$ =
5 and 10, and the results are shown in Figure 8. Bimodal (conjugate) datasets for these
dispersed and sparse patterns lie across the 1:1 line on the fabric plot (Figure 8a; $\kappa$ = 5 in
blue, $\kappa$ = 10 in yellow). Quadrimodal datasets for these parameters are also noisy, with some
fabrics lying below the 1:1 line (Figure 8b; $\kappa$ = 5 in blue, $\kappa$ = 10 in yellow). Under these
conditions of low $\kappa$ (dispersed) and low $n$ (sparse), it can be difficult to separate bimodal
(conjugate) from quadrimodal fault patterns. However, we assert that this may not matter: a
noisy and disperse 'bimodal' conjugate fault pattern is in effect similar to a polymodal pattern
i.e. slip on these dispersed fault planes will produce a bulk 3D triaxial strain.



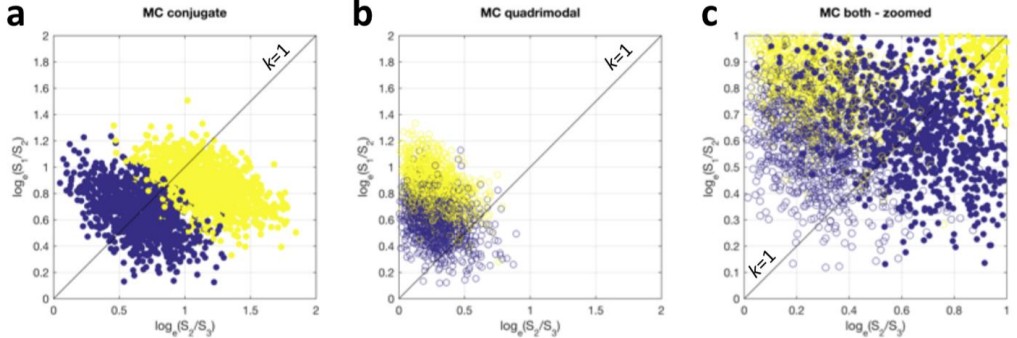


**Figure 8.** Eigenvalue ratio plots of synthetic data to illustrate the impact of dispersion on the ability of this plot to discriminate between conjugate (bimodal) and quadrimodal fault data. **a**) Monte Carlo ensemble of 2000 conjugate fault populations (mixtures of two equal Watson distributions), with $\kappa$ varying from 5 (dark blue) to 10 (yellow). **b**) Monte Carlo ensemble of 2000 quadrimodal fault populations (mixtures of four equal Watson distributions), with $\kappa$ varying from 5 (dark blue) to 10 (yellow). **c**) Data from a) and b) merged onto the same plot and enlarged to show the region close to the origin. Note the considerable overlap between the conjugate (bimodal) data with the quadrimodal data, especially for $\kappa = 5$ (dark blue).

To assess the relative performance of the two tests presented in this paper, we generated synthetic bimodal and quadrimodal distributions and compared the resulting p-values from applying both the $S_1$-$S_3$ and $S_{11}$-$S_{33}$ tests to the same data. The results are shown in Figure 9, displayed as cross-plots of $p(S_1$-$S_3)$ versus $p(S_{11}$-$S_{33})$. While there is a slight tendency for the p-values from the $S_{11}$-$S_{33}$ test to exceed those of the $S_1$-$S_3$ test (i.e. the points tend on average to plot above the 1:1 line), neither of the tests can be said to 'better' or more 'accurate'. We therefore recommend the $S_1$-$S_3$ test as simpler and sufficient.

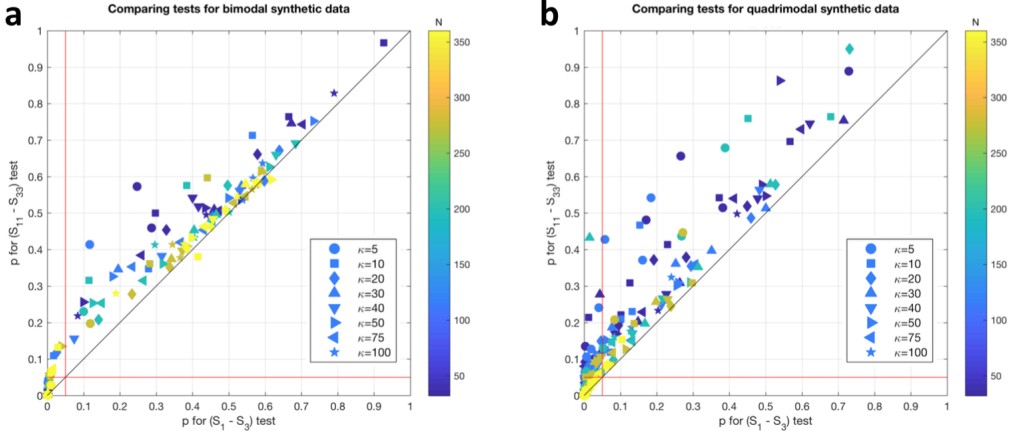


**Figure 9.** Eigenvalue ratio plots comparing the relative performance of the two tests proposed in this paper. The red lines denote p-values for either test at p=0.05, and the diagonal black line is the locus of points where $p(S_1$-$S_3) = p(S_{11}$-$S_{33})$. **a**) For bimodal synthetic




datasets with size (N) varying from 32-360 and concentration ($\kappa$) varying from 5-100, both
tests perform well and reject the majority of the datasets ($p \gg 0.05$). The p-values for the $S_{11}$-
$S_{33}$ test are, on average, slightly higher than those for the $S_1$-$S_3$ test across a range of dataset
sizes and concentrations. **b**) For quadrimodal synthetic datasets, many of the p-values are <
0.05, and this especially true for the larger datasets (higher N, green/yellow). Smaller datasets
(blue) can return p-values > 0.05.

## 6. Summary

Bimodal (conjugate) fault patterns form in response to a bulk plane strain with no extension
in the direction parallel to the mutual intersection of the two fault sets. Quadrimodal and
polymodal faults form in response to bulk triaxial strains and probably constitute the more
general case for brittle deformation on a curved Earth (Healy et al., 2015). In this
contribution, we show that distinguishing bimodal from quadrimodal fault patterns based on
the orientation distribution of their poles can be achieved through the eigenvalues of the 2nd
and 4th rank orientation tensors. We present new methods and new open source software
written in R to test for these patterns. Tests on synthetic datasets where we controlled the
underlying distribution to be either bimodal (i.e. conjugate) or quadrimodal (i.e. polymodal,
orthorhombic) demonstrate that a combination of fabric eigenvalue (modified Flinn) plots
and our new randomisation tests can succeed. Applying the methods to natural datasets from
a variety of extensional normal-fault settings shows that 5 out of the 6 fault patterns
considered here are probably polymodal. The most tightly constrained natural dataset
(Gruinard) displays clear orthorhombic symmetry and is unequivocally polymodal. We
encourage other workers to apply these tests to their own data and assess the underlying
symmetry in the brittle fault pattern and to consider what this means for the causative
deformation.

## Acknowledgements

DH gratefully acknowledges receipt of NERC grant NE/N003063/1, and thanks the School of
Geosciences at the University of Aberdeen for accommodating a period of study leave, during
which time this paper was written.

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

Geological Society, London, Special Publications, 439.