# Peer review of "Bimodal or quadrimodal? Statistical tests for the shape of fault patterns"

_Solid Earth, 2018_

## Referee Comment (RC1) · Anonymous Referee #1 · 11 Jun 2018

I have now revised with pleasure the manuscript by Healy and Jupp on statistical treatment of bimodal and quadrimodal fault patterns formed in response to a single tectonic event.

While the problem is surely addressed in a correct way from a statistical point of view (and I am not an expert of statistical treatments), my concerns regard the geological significance of results obtained by Healy and Jupp and t5hei application to geological processes of faulting.

(1) First of all, the typical geological contexts/ processes leading to the formation (in a single event or in a relatively shor7 time) of bimodal and quadrimodal/polymodal fault patterns should be explained and investigated in the first/introductory sections of the paper, and then reconsidered in the discussion and conclusive sections.

[Figure]

(2) Once these contexts/ processes are explained, natural fault patterns to be statistically tested should be taken from these explicit cases or, for comparison/contrast, from different cases.

(3) In the case of quadrimodal/polymodal fault patterns, I do not see many alternative cases (I might be wrong) to polygonal faults that are polymodal (normal) faults developed in one single event. I know that many of these faults are known only from offshore areas thanks to seismic images. I wonder whether it would be possible a statistical test using only the fault strikes (instead of fault attitude) that are documented in many papers on polygonal faults based on seismic data. It is also true, however, that polygonal faults start to be known and measured also in many inland cases. For references on papers on offshore and onshore polygonal normal faults I refer the authors to the following paper (Wrona et alii 2017): https://www.frontiersin.org/articles/10.3389/feart.2017.00101/full

In synthesis, I suggest the authors to consider fault data from polygonal normal faults and to better explain the geological relevance of their statistical results and applications to geological processes.

---

## Referee Comment (RC2) · Anonymous Referee #2 · 11 Jun 2018

Dear editor and authors, This is a technical paper that present a new numeric tool for the analysis of fault/fracture sets. As the authors discuss, it not always easy to separate large (or small) orientation databases into sub-sets, and to identify the type of distribution. Therefore, the proposed method has the potential to be utilized by researchers in the field. In this respect, the paper can be a good contribution. The paper is clearly written and include multiple clear illustrations of synthetic and field data. It can be shorten as suggested below. The authors are strongly urged to revise the paper following the suggestions below. Major comments 1. The paper is written as a statistical manuscript and not as a tool for geologists. The methodology is presented as a black box without sufficient explanation on the rationale behind it and/or the statistical terminology. Below are few examples. a. The "eigenvalues of the 2nd and 4th rank orientation tensors"

[Figure]

have relations to the actual distribution of the orientation data. The authors MUST give the eigenvalues AND the associated eigenvectors of the idealized cases of Figs. 1 d-f. b. The R language was probably chosen due to its power in statistical calculations, but the link to the code (lines 85-86) does not work and the potential users MUST get a compiles code. c. The paper presents the calculations results in relative length, with almost no discussion of the geologic significance. 2. It is not clear why there is a need for 16 synthetic sets (Fig. 3, 4), it appears as an exercise in statistics rather than a tool for geologists. Cut to 6 synthetic sets. This will also shorten the paper. 3. Lines 200-216 are the key for understanding the rationale of the method, but the authors just describe Fig. 6 without explaining the PHYSICAL meaning of the eigenvalues of S1, S2 and S3 and their relations. For example, Fig. 6 is a modified Flinn diagram is a presentation of the shape of strain ellipsoid by displaying the relations between strain axes of the ellipsoid. Such links to geology will strengthen the utility of the paper. 4. The paper deals only with the orientations of the fault surfaces, while the proposed method can be applied to other structural elements in geology. For example, the slip directions along faults that are essential for stress inversion and strain inversion of fault data, orientations of cross-bedding in sandstone deposits, and the orientations of joint sets (for separation of extension phases). This limitation by the authors simplifies the analysis, but restricts its utilization. It is suggested that the authors discuss the other cases of oriented data and maybe suggest possible utilization by the proposed method.

———————————————————————

---

## Author Comment (AC1) · 9 Jul 2018

We thank the reviewer for their concise and constructive comments on our manuscript. We address the issues raised in sequence in the text below, complete with any explicit changes we have made to our manuscript.

1. Geological context and processes of bimodal and quadrimodal fault patterns should explained in the introductory sections and then reconsidered In the discussion and conclusion.

Reply: we disagree. Full reference is made to relevant papers that discuss the key differences between bimodal/conjugate and quadrimodal/polymodal fault patterns. Our manuscript describes a new method to distinguish between these distinct patterns, and

we think a full repetition of the issues is not warranted. We do highlight the key issues in the Introduction (section 1), and address the issues raised by our statistical analysis in the Discussion (section 5).

2. Once these contexts/ processes are explained, natural fault patterns to be statistically tested should be taken from these explicit cases or, for comparison/contrast, from different cases.

Reply: this is exactly what we do; in addition to the synthetic datasets built from Watson distributions, we use published datasets of natural normal fault orientations previously ascribed to either bimodal/conjugate origin (e.g. Peacock & Sanderson, 1992) or to quadrimodal (Krantz, 1989) patterns.

3. In the case of quadrimodal/polymodal fault patterns, I do not see many alternative cases (I might be wrong) to polygonal faults that are polymodal (normal) faults developed in one single event. I know that many of these faults are known only from offshore areas thanks to seismic images. I wonder whether it would be possible a statistical test using only the fault strikes (instead of fault attitude) that are documented in many papers on polygonal faults based on seismic data. It is also true, however, that polygonal faults start to be known and measured also in many inland cases. For references on papers on offshore and onshore polygonal normal faults I refer the authors to the following paper (Wrona et al., 2017).

Reply: polygonal faults remain enigmatic, and in comparison to bimodal or quadrimodal fault patterns they are statistically insignificant. A quantification of fault strikes from polygonal arrays has already been performed e.g. in the cited paper by Wrona et al., 2017. Our method to distinguish between bimodal and quadrimodal fault patterns fundamentally depends on the input of fully 3D orientation data – i.e. the poles to the fault planes – and not just the fault strikes.

---

## Author Comment (AC2) · 9 Jul 2018

We thank the reviewer for their concise and constructive comments on our manuscript. We address the issues raised in sequence in the text below, complete with any explicit changes we have made to our manuscript.

1. The paper is written as a statistical manuscript and not as a tool for geologists. The methodology is presented as a black box without sufficient explanation on the rationale behind it and/or the statistical terminology. Below are few examples. a. The "eigenvalues of the 2nd and 4th rank orientation tensors" have relations to the actual distribution of the orientation data. The authors MUST give the eigenvalues AND the associated eigenvectors of the idealized cases of Figs. 1 d-f. b. The R language was

probably chosen due to its power in statistical calculations, but the link to the code (lines 85-86) does not work and the potential users MUST get a compiles code. c. The paper presents the calculations results in relative length, with almost no discussion of the geologic significance.

Reply: our methodology is definitely NOT a 'black box'! We present the underlying equations AND the source code for our software. a) The idealised cases in Figures 1d-f are just schematic, as noted in the caption; b) The link to the code has been corrected and now works, together with a new user guide; c) Our ms contains over 2 sides of Discussion of the results, centred on their geological significance.

2. It is not clear why there is a need for 16 synthetic sets (Fig. 3, 4), it appears as an exercise in statistics rather than a tool for geologists. Cut to 6 synthetic sets. This will also shorten the paper.

Reply: we disagree, strongly, on this point. The synthetic datasets have been carefully chosen to cover a range of cases to mirror the spread of natural datasets. We vary kappa, the concentration parameter in the Watson distribution, and n, the size of the dataset, for both bimodal and quadrimodal distributions. Statistical rigour demands that we test our method across the expected span of natural datasets.

3. Lines 200-216 are the key for understanding the rationale of the method, but the authors just describe Fig. 6 without explaining the PHYSICAL meaning of the eigen-values of S1, S2 and S3 and their relations. For example, Fig. 6 is a modified Flinn diagram is a presentation of the shape of strain ellipsoid by displaying the relations between strain axes of the ellipsoid. Such links to geology will strengthen the utility of the paper.

Reply: there seems to be a misunderstanding about Figure 6. It is indeed a modified Flinn plot after Ramsay (1967), showing the ratios of the eigenvalues of the orientation tensor (or matrix). Flinn and Ramsay were concerned with the eigenvalues of the strain tensor. There may be a relationship between the strain tensor and the orientation

tensor of a faulted region but that is beyond the scope of our manuscript.

4. The paper deals only with the orientations of the fault surfaces, while the proposed method can be applied to other structural elements in geology. For example, the slip directions along faults that are essential for stress inversion and strain inversion of fault data, orientations of cross-bedding in sandstone deposits, and the orientations of joint sets (for separation of extension phases). This limitation by the authors simplifies the analysis, but restricts its utilization. It is suggested that the authors discuss the other cases of oriented data and maybe suggest possible utilization by the proposed method.

Reply: we present our new method with a focus on discriminating between bimodal and quadrimodal fault patterns. We emphatically do NOT restrict it's utilisation for other applications: in contrast, by providing an open access manuscript and open source R code we are ENABLING application to other domains. Moreover, the precise nature of the suggested application of our method to these other domains (slip directions, cross-bedding, and joint sets) remains unclear. In terms of fault patterns, we are addressing a problem in the underlying symmetry (or lack thereof) in the orientation distribution, and the implications this has for the mechanics of brittle failure.

---

## Referee Report (RR1)

**Comments on SE ms. by Healy and Ruup**

231 $\mu$ is often used for the friction coefficient: change it to something else

235 is gamma intersection angle? If so, point it out.

249 and 252 S and T are commonly used for the principal stresses.

279 tests of bimodality and quadrimodality appears to be fine and merit the publication of the manuscript.

318-324 Good discussion of the Chimney Rock data set, a detail of which is hard to find in any place in the literature. I don't have anything to say about the data from central Italy attributed to Roberts, 2007. Sorry, I am too lazy to dig it out!

372 The word "probably" is unnecessary.

374-376 The main results are worthy in a statistical sense as I mentioned above. Perhaps that is why the authors find a better fit with the synthetic datasets. Given that most map-scale natural faults evolve or grow by interaction, splaying, coalescence, and in some cases, reactivation under progressive material and stress rotation, naturally they are much more complex (Aydin and Zhong; Rock Fracture Knowledgebase). I don't expect the authors discuss all of these mechanisms in one manuscript, but they may just point out why most natural faults are more complicated.

By the way, I like the interactive review system and I don't mind my name and comments being open to the authors and the public.

Atilla Aydin

aydin@stanford.edu

---

## Author Response (AR2)

**Response to Referee #3**

We thank the reviewer for their concise and constructive comments on our manuscript. We address the issues raised in sequence in the text below, complete with any explicit changes we have made to our manuscript.

*This is a short, straightforward paper addressing an important issue in structural geology. The statistical tests derived here should be of wide interest and applicability.*

*My maths is too rusty to follow all the details of the statistical treatment. What I can follow looks right and the results are qualitatively sensible.*

We thank the reviewer for his belief that this work should be of wide interest and applicability.

*My only comments on the paper concern nomenclature of the patterns of poles to faults. I realise that the nomenclature is the published one of Healy et al. (2016) but the comments still need consideration.*

*The issues arise in lines 43 to 72 and in Figure 1.*

*a) The term 'bilateral' is used without definition in line 72, presumably to mean a pattern that is bilaterally symmetrical about one or more of the principal planes of the distribution. The issue is that all the illustrated ideal patterns are bilaterally symmetric about all the three principal planes, so 'bilateral' doesn't seem to discriminate any one pattern.*

*b) The term 'polymodal' is applied to a pattern that is actually bimodal, with each mode dispersed along a small circle about sigma1. Another term, like 'dispersed bimodal' seems more accurate.*

*c) The term 'orthorhombic' is being equated to the 'polymodal' pattern only? Actually all the illustrated ideal patterns have orthorhombic symmetry, as they must have if they have three orthogonal mirror planes.*

*I suppose the central point is that the number of modes in a distribution is unrelated to its symmetry. A distribution needs both descriptors to specify it. So distributions could be bimodal and monoclinic, or polymodal and triclinic, or many other variants. Your statistical method (I think) only applies to orthorhombic distributions and maybe this should be stated up front.*

*This lack of clarity might cause confusion in the future literature?*

In line 72, we now make this more explicit to address points a)-c) above. However, note that using 'polymodal' for the type of distribution shown in Figure 1f is correct if (and this is where our work is relevant) there is no central tendency within each of the clusters i.e. it is literally 'many moded', rather than a noisy bimodal pattern. We have added text to the abstract to highlight the restriction to orthorhombic symmetry.

We hope the changes we have made will serve to reduce the chances for confusion.

**Response to Referee #4**

We thank the reviewer for their concise and constructive comments on our manuscript. We address the issues raised in sequence in the text below, complete with any explicit changes we have made to our manuscript.

*231 $\mu$ is often used for the friction coefficient: change it to something else*

Our paper is not about mechanics or frictional sliding; we doubt there will be any confusion with using mu for this term. Left as is.

*235 is gamma intersection angle? If so, point it out.*

Explained on line 236.

*249 and 252 S and T are commonly used for the principal stresses.*

The S notation for the eigenvalues comes from Woodcock (1977); we doubt if there will be confusion with stresses. Left as is.

*279 tests of bimodality and quadrimodality appears to be fine and merit the publication of the manuscript.*

Good, thanks!

*318-324 Good discussion of the Chimney Rock data set, a detail of which is hard to find in any place in the literature. I don't have anything to say about the data from central Italy attributed to Roberts, 2007. Sorry, I am too lazy to dig it out!*

Thanks!

*372 The word "probably" is unnecessary.*

Yes, deleted.

*374-376 The main results are worthy in a statistical sense as I mentioned above. Perhaps that is why the authors find a better fit with the synthetic datasets. Given that most map-scale natural faults evolve or grow by interaction, splaying, coalescence, and in some cases, reactivation under progressive material and stress rotation, naturally they are much more complex (Aydin and Zhong; Rock Fracture Knowledgebase). I don't expect the authors discuss all of these mechanisms in one manuscript, but they may just point out why most natural faults are more complicated.*

Added new lines in this section.  Thanks.